# Unraveling spatial cellular pattern by computational tissue shuffling

Elise Laruelle [1], Nathalie Spassky [1✉] & Auguste Genovesio [1✉]

Cell biology relies largely on reproducible visual observations. Unlike cell culture, tissues are heterogeneous, making difficult the collection of biological replicates that would spotlight a precise location. In consequence, there is no standard approach for estimating the statistical significance of an observed pattern in a tissue sample. Here, we introduce SET (for Synthesis of Epithelial Tissue), a method that can accurately reconstruct the cell tessellation formed by an epithelium in a microscopy image as well as thousands of alternative synthetic tessellations made of the exact same cells. SET can build an accurate null distribution to statistically test if any local pattern is necessarily the result of a process, or if it could be explained by chance in the given context. We provide examples in various tissues where visible, and invisible, cell and subcellular patterns are unraveled in a statistically significant manner using a single image and without any parameter settings.

[1] Institut de Biologie de l'Ecole Normale Supérieure (IBENS), CNRS UMR8197, INSERM U1024, PSL Research University, 46 rue d'Ulm, 75005 Paris, Paris, France. ✉email: nathalie.spassky@ens.psl.eu; auguste.genovesio@ens.psl.eu

Since the advent of high throughput dispensing and auto-mated microscopy, methods and software tools for large-scale single-cell image analysis have blossomed and enabled profiling and comparison of large ranges of perturbations on cell cultures[1–4]. Variance estimation and statistical testing in these approaches are achieved by simply producing several replicates per condition. This is made possible by the fact that cell culture permits robust standardized and sometimes fully automated replication of a sample condition. In contrast, while it is still possible to detect and quantify a single-cell event in large slide of cell tissues, their comparison and statistical analysis have remained hampered, apart from stereotypic exceptions, by the imprecision of microdissection, spatial inhomogeneity of samples, and notorious replicate variability[5]. In fact, spatial inhomogeneity is what makes tissues interesting to study in comparison to cell culture that often spreads a single or a few cell types uniformly but barely matches the spatial organization of these cells in an organism. In tissue samples, the state of a cell within its local context is observed once, and this exact context can in general barely be reproduced with precision. Therefore, as heterogeneity across a sample is the rule, obtaining robust standardized repli-cates is difficult for a single-cell event and impossible for a small cell patch or cell organization pattern observed locally. The unavailability of reliable replicates of such an event makes com-parison of between-versus-within group variance irrelevant and statistical evidence unworkable.

However, what is sought in studies that relies on tissue sample observation are factors underlying cell organization, develop-mental process, or disease progression, independent of the variability between observations. From this point, how to deal with the impossibility of obtaining robust standardized replicates of an event? Is it possible to statistically test for the existence of a local cell to cell relationship from a single replicate? How to assess the existence and detect the heterogeneity of a local phenotype across one tissue sample? All these questions can be summarized in one: is the cell organization, observed at a specific location, driven by molecular or mechanical factors, or is it likely to be expected by chance given the distribution of cell shape and size? Being able to systematically answer this question is of growing importance as bridging the gap between profiles of single-cell gene expression and spatial cell relationships and morphology in tissue samples is at reach[6,7]. Studying further cell communication and organization will require the ability to decipher non-random spatial relationships between cells, in order to complement and possibly explain single-cell gene expression.

While modeling of tissues is a rich area, statistical methods to study the spatial organization of actual cell images, especially of tissues, were not extensively discussed in the literature[8]. Marked point process statistics offer compelling tools to study spatial autocorrelations but, as their name suggests, they become inap-propriate for compartments, especially with irregular shapes such as cells[9]. In recent years, deep learning approaches were mostly successful for classification problems where they provided state of the art results, especially with cell culture, as data acquisition can be properly standardized[10,11]. These approaches were also applied to tissue mainly for diagnostic purposes, but the quality of the results remained highly dependent on the possibility and the quality of standardization[12]. More importantly, deep learning approaches do not provide solutions to infer if a local piece of tissue is structured or not from a single image, which is needed to decipher processes from heterogeneous tissue samples in basic research in cell biology. Altogether, there is no method providing a systematic way to assess, in a statistically relevant manner, if a local cell to cell relationship or local spatial pattern can be observed by chance given the cells or if it is likely due to a molecular underlying process.

A reason that could explain this current lack of a reference method could be that cells in a tissue are heterogeneous and their shape and size can vary extensively across tissue samples. The last makes the null distribution of any cell relationship-based feature difficult to construct either empirically using randomization approaches or analytically because the spatial arrangement of various cell sizes and shapes is difficult to model. Indeed, as we shall see further, size and shape of the cells do influence their local spatial arrangement and thus need to be preserved in the con-struction of the null distribution.

In this work, we present an approach that brings a robust and broadly applicable computational solution to this issue. We first introduce SET (for Synthesis of Epithelial Tissue), an accurate modeling of the cell tessellation observed in an image of tissue. This model is first used to reconstruct a close approximation of the original image's cell tessellation (called "reconstruction by SET" in the following) and further used to generate synthetic alternative tessellations made of the same cells located randomly (called "random SET"). Generating thousands of synthetic images using random SET enabled us to build a null distribution for any quantitative features extracted from a local group of cells. We demonstrate that computing the same quantitative feature from the reconstruction by SET provides an accurate $p$ value to accept or reject the null hypothesis corresponding to a random appari-tion of the pattern. Our method does not require any input parameters, making it a stable and easy to use tool for analysis with no need for ad hoc parameter setting or fine tuning. Fur-thermore, it requires only one image to draw statistically relevant conclusions. A compelling advantage given an observed position in a tissue, in general, can hardly be defined exactly such that to be observed repeatedly in several replicates or in mutants. Interestingly, it allows us to independently investigate various locations in a single piece of tissue to study its heterogeneity. We illustrate the broad relevance of this approach with various examples of epithelia in several organisms.

## Results

**SET: a model to reconstruct epithelial cell tessellation.** We first wondered how the tessellation drawn by the cell boundaries of an epithelium could be modeled and then accurately reproduced artificially from a minimum set of parameters per cell. We then developed an approach that takes as input a segmented cell image of tissue, as many software packages now offer rather precise cell segmentation[13–19] (Fig. 1a). We then defined a specific flexible parametric distance function that could be fit such that the level one of the distance map matches the contour of the cell (Fig. 1b, Supplementary movies 1–5 and "Methods"). After fitting, this parametric distance function combines eight parameters per cell. It included three positional parameters for each cell: the location of the cell $x$, $y$, and the angle alpha formed by the principal axis of the cell and the $x$-axis. It also included five parameters describing the shape of each cell: the lengths of its two principal axes $s_1$, $s_2$, two parameters $a_1$ and $a_2$ accounting for the asymmetry of the cell and a last parameter $p$ that allows for a cell to possibly include corners. Once fitted, our approach uses these single-cell-dependent metrics to compute a tessellation by iteration with a modified Lloyd algorithm (Fig. 1c and "Methods"). The Lloyd algorithm enables to adjust cells to its neighbors and feel gaps between them as it associates all pixels of an image to the closest cell in a locally optimal way. At initialization, the five shape parameters for each cell remained fixed along the process to preserve cell shapes. However, the positional parameters of each cell were let free to evolve and all the pixels in the plane were subject to an iterating assignment process until convergence. Applying the algorithm to an image by initializing the positional

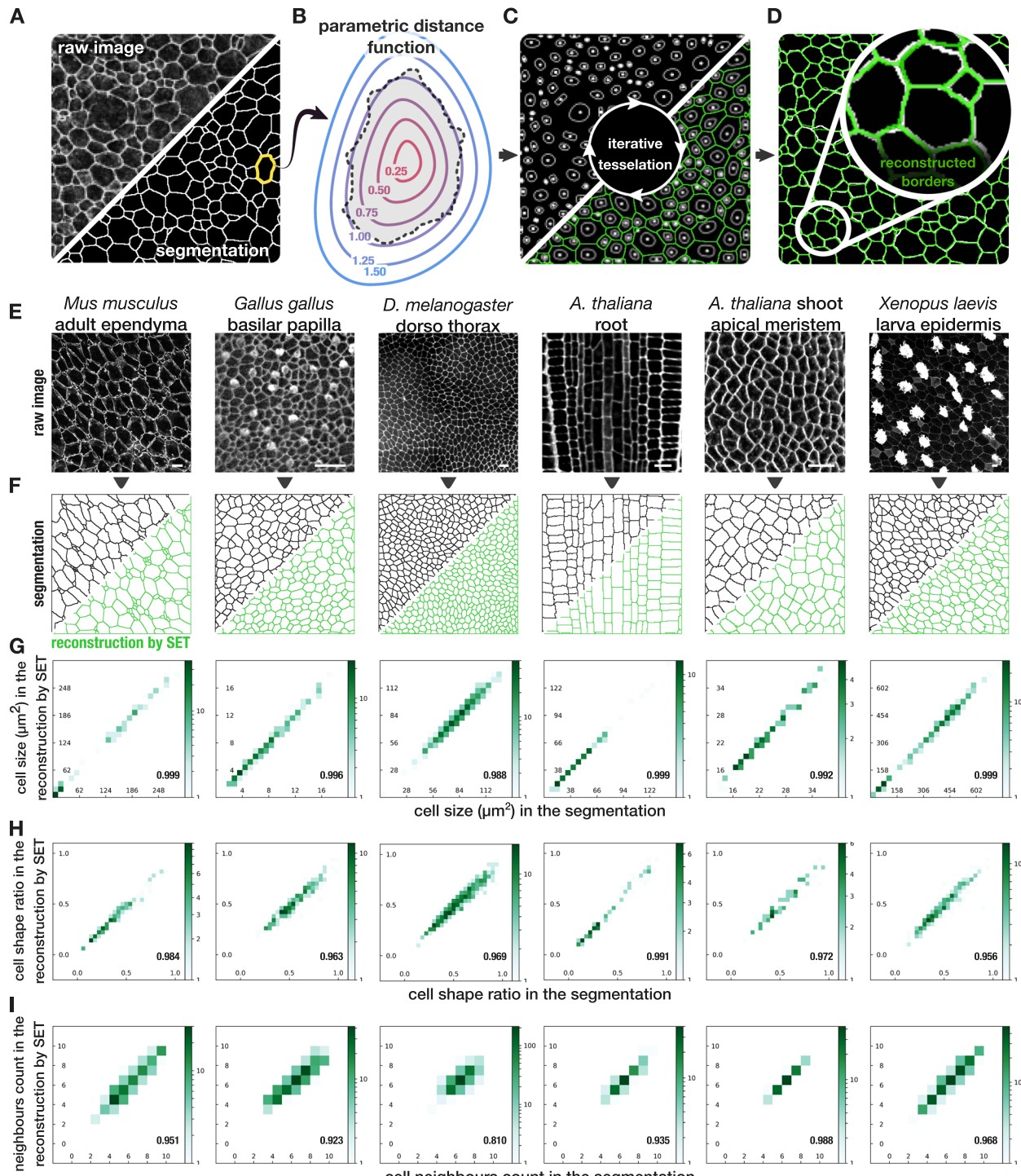

**Fig. 1 Reconstruction by SET: parametric modeling and accurate reconstruction of an epithelial cell tessellation. a–d** Flowchart of the SET method. **a** Top left, an image of P1 mouse ependyma in which cell junctions were labeled with ZO1 antibody (Supplementary Fig. 1) and, bottom right, the segmentation of all cells in this image. **b** Fitting of the level one of AMAT distance function on each cell such that each pixel on the cell border is approximately at a distance 1 from the cell center. **c** Starting from the initial cell positions and using all AMAT distance functions previously defined, an iterative process (Lloyd-like algorithm) is used to affect all pixels to the closest cell until convergence. **d** Visualization of the reconstruction by SET in green over the original segmentation in white. **e** Raw images of various epithelia with marked cell borders (membranes or cell walls, see "Acknowledgements", "Methods", and further results for images ressources), scale bar: 10 μm. **f** For each panel, top left in gray is the segmentation of the image in **e** and, bottom right in green is the corresponding reconstruction by SET. **g–i** Comparison of quantitative features computed on each single cell from segmented tissue and on the corresponding cell in the reconstruction by SET. Pearson coefficient correlation calculated on all cells except the border cells (as they are incomplete). **g** Cell size. **h** Ratio of the minor axis to the major axis of the cell. **i** Number of cell neighbors. For all plots $N = 1$ image.

parameters, with the original values obtained after detection, enabled us to precisely reconstruct the cell tessellation by SET (Fig. 1d and "Methods"). To ensure our model could accurately and genuinely reproduce various cell tissues, we applied it to adult mice ependyma, *Drosophila* thorax epithelium, *Xenopus* epithelium, Chick Basilar Papilla, root and shoot apical meristem of *Arabidopsis thaliana* (Fig. 1e). Features computed at the single-cell level, such as the cell area or shape ratio, could be closely reproduced in the reconstruction by SET (Fig. 1g, h). Moreover, the cell neighbor count, a feature at the cell organization level, also showed a high correlation between the reconstructions by SET and the original segmentations in all provided examples, assessing that cell neighborhood was also maintained (Fig. 1i).

**Relocating cells by SET preserves single-cell properties**. The model previously described can be considered to be a parametric function characterizing a given image of tissue by a set of parameters describing cells position and shape. Modifying those parameters could then produce alternative artificial tissues. In the following, we altered only the initial positional parameters of the cells to make possible the construction of a tessellation made of the exact same cells with the same shape as in the original image but reorganized differently (Fig. 2a and "Methods"). Following the cell displacement, it was furthermore possible to morph the content of each original cell to its new location using barycentric coordinates. Briefly, barycentric coordinates enable it to represent any point of the plane as a weighted sum of a set of anchor points rather than by Cartesian coordinates. Modifying these anchor points enables it to smoothly displace all points thus defined ("Methods" and Supplementary Fig. 2). The initialization of positional parameters can be deterministic (alternative SET) as the last two panels of Fig. 2a demonstrate or random (random SET) as in all the following examples in Fig. 2b generated from the examples provided in Fig. 1e. In both cases, this new tessellation preserved single-cell features as cell area or shape ratio (Fig. 2c, d and Supplementary Fig. 3) and as expected broke neighborhood organization (Fig. 2e versus Fig. 1i and Supplementary Fig. 3). Generating random SET of the observed tissue showed cell patterns that could have been obtained with these exact cells if their location were to be distributed randomly. The result is an algorithm that can generate thousands of synthetic images, all containing the exact same cells as a single real image, that will subsequently be used to compute the null distribution of any quantitative feature. Importantly, the algorithm is robust and does not require any input parameter to be set or fine-tuned. All cell parameters are automatically estimated from the cell segmentation. Interestingly, because the synthesized cell shape is by construction very close to the original one, the cell content could also be transported smoothly without significant distortion, preserving organelles location (Fig. 2a). Therefore, this approach enabled us to also study the cell to cell relationships of intracellular organelles, as demonstrated further.

**SET spatial shuffling unveils patterns in the adult ependyma**. We first used our approach to study cell organization in the neurogenic niche of the adult mouse brain. This niche is retained in the walls of the adult lateral ventricles (LV) and is composed of neural stem cells clustered in the center of multiciliated ependymal cells, described as pinwheel-like structure (Fig. 3a)[20]. The apical surface of the two cell types are 10-fold size-asymmetric. Cells with a large apical surface are multiciliated ependymal cells while cells with a small apical surface are stem cells. However, with such a size asymmetry between cell types and the subtlety of this phenotype, observing such a structure could possibly be due to chance. To test this hypothesis, we built a null distribution

from one single image by counting the number of stem cells in direct contact with another stem cell in the reconstruction by SET than in a thousand random SET. The number of stem cells touching another stem cell in the distribution of random SET was systematically smaller than in the reconstruction by SET (Fig. 3c). It indicated that the null hypothesis stating a random positioning of stem cells could safely be rejected with a $p$ value that was at least $10^{-3}$. Accordingly, multiciliated-multiciliated cell contacts were found higher in the random SET than in the reconstructed SET (Supplementary Fig. 4B). To check the consistency of our approach, we also performed the same test on another image obtained from approximately the same location in a replicate individual of the same age and results were confirmed (Supplementary Fig. 5). Thus, while nonobvious by eye, our method concludes without the need of a control and with a single image that the stem cells cluster in the adult ependym through an active mechanism.

In order to evaluate the relevance of the method we propose, we compared it to two other approaches: a shuffle of the same ratio of stem cells on an hexagonal grid and a shuffle of the same ratio of stem cells on the original image segmentation (see "Methods" for details and Supplementary Fig. 6). Intriguingly, both of these approaches produced null distributions that contradicted our results, concluding that stem cells were unclustered (Fig. 3c and Supplementary Fig. 5). To clear this up, we generated three alternative SET with known artificial patterns representing different clustering configurations: an image engineered such that stem cells obviously cluster (Fig. 3d), an image with random positioned stem cells (Fig. 3f), and an image engineered such that stem cells are isolated from one another on purpose (Fig. 3h). Those SET were all obtained from the same original image (Fig. 3a) where the stem cells only were artificially moved. Accordingly, ependymal cells were initialized at their original location while stem cells were initialized to manually set locations, then the model was let free to autoadjust all cells together. This way, both cell shape and cell type ratio distributions were preserved in the three engineered artificial examples. The results show that the two other methods did not succeed in detecting an obvious clustering of cells (Fig. 3e). Furthermore, they also show that a random sampling of stem cell locations was interpreted by the two other approaches as a repelling of cells (Fig. 3g). The three methods agreed only to detect a true repelling configuration of the cells (Fig. 3i). Altogether, these results indicate that our approach is the only one among these three that could correctly detect all known configurations. It also shows that the two other approaches are strongly biased, leading to possibly false interpretation. Whether the local spatial arrangement of cells is random or not, we figured out using additional SET simulations and real images that the cell neighbors count is related to the distribution of their size and shape in a hardly predictable fashion (Supplementary Fig. 7). Therefore, these single-cell properties must be considered in the construction of a correct null distribution to test the significance of an observed cell neighborhood pattern. In opposition to SET, shuffling in a hexagonal grid considers fix and round cells and shuffling in the actual segmentation disregards the relationship between cell type, and cell size and shape, which produces the same neighbor count null distribution for both subpopulations as demonstrated by further simulations (Supplementary Fig. 8). While biased in the presented case, these other approaches could be valid in the case where cells are all similar in size and shape, a hypothesis that is rarely met.

**SET partial shuffling uncovers subpopulations patterns**. We then extended the approach to tissues that contain more than two

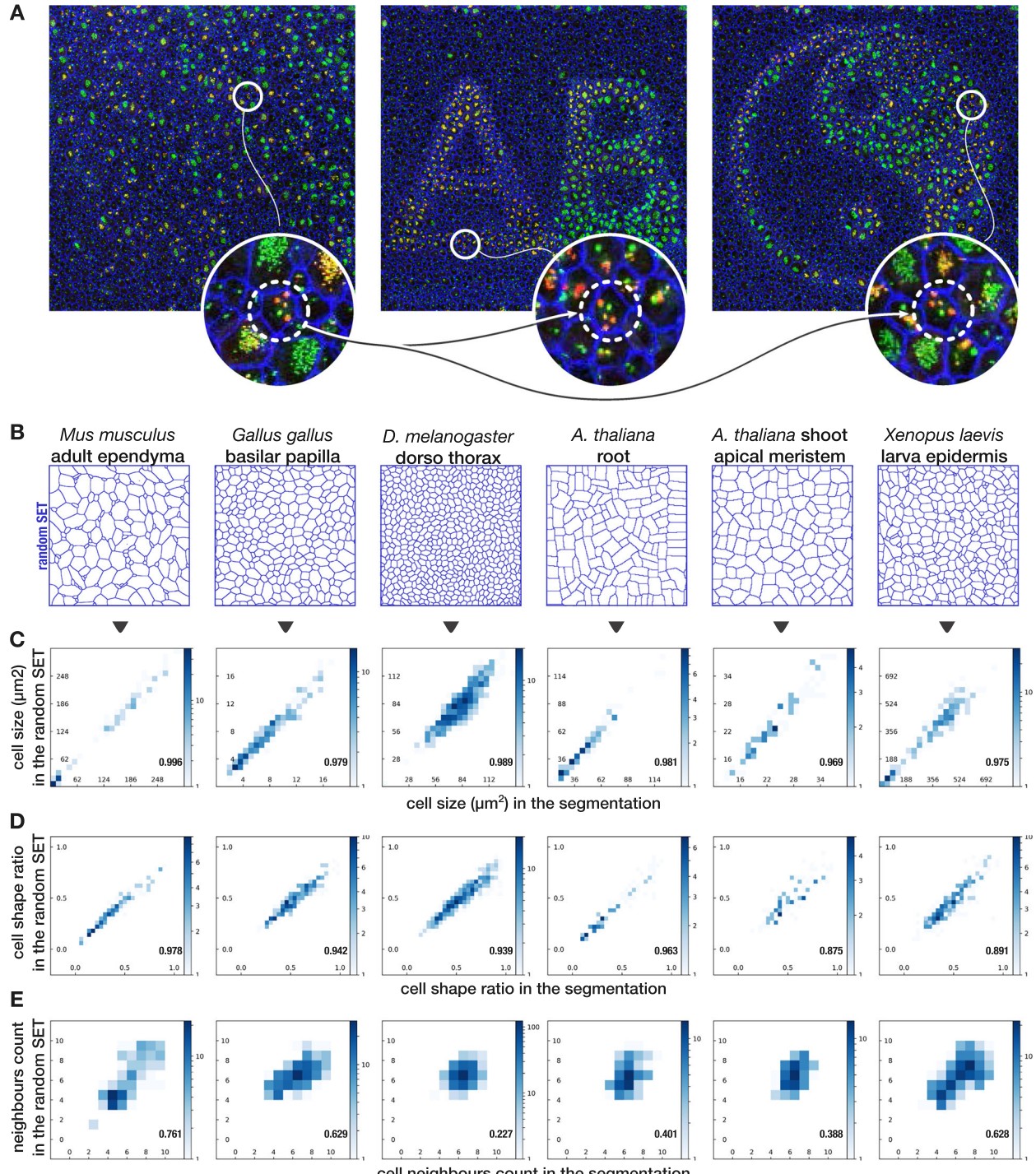

**Fig. 2 Generation of alternative SET preserves single-cell size and shape ratio while expectedly breaks local cell organization. a** Left: an image of P3 mice lateral ventricular surface where borders (ZO1 antibody, blue), centrioles (Centrin2-GFP mice, green), and procentrioles (Sas6 antibody, red) are labeled. Middle and right: two alternative SET with the exact same 2822 cells taken from **a** and artificially moved to specified locations for illustrating the capability of SET. Only the positional parameters are modified so that cell shapes remain approximately the same. All cells were moved except those touching the borders of the image (as they are incomplete). Zoom on the same cell in the three images shows that intracellular content of each cell could also be transported to its new location using barycentric coordinates relative to the cell contour (see "Methods" for details). Interestingly, single-cell properties such as size and cell minor axis to major axis ratio is preserved while neighborhood is expectedly broken (Supplementary Fig. 3). **b** The same process is applied on all images presented in Fig. 1e but this time, location and orientation are randomly initialized. **c–e** 2D histograms showing the comparison of quantitative features computed for each cell both on the segmentation and on the random SET. Pearson correlation coefficient computed on all the cells except the border cells: **c** Cell size correlation. **d** Cell minor axis to the major axis ratio correlation. **e** Cell neighbors count correlation. The correlation of this last feature is expectedly broken as it depends on the surrounding of each cell which is randomly shuffled by the process. For all plots $N = 1$ image.

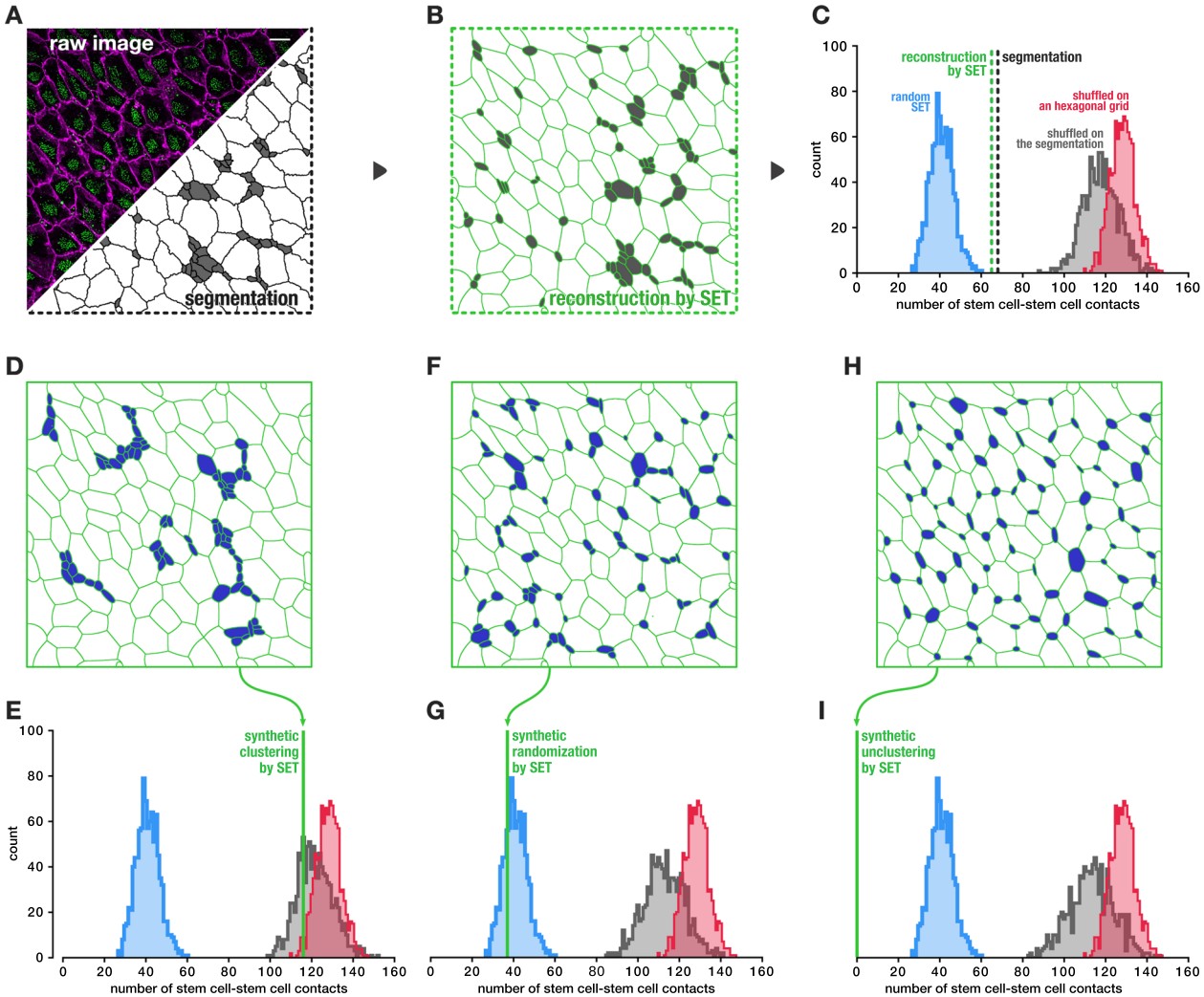

**Fig. 3 Stem cells are organized in clusters surrounded by ependymal cells along the ventricular surface of the lateral ventricles, comparison with other approaches. a** Top left, image of an en face view of the lateral ventricular surface of a P30 labeled with ZO1 antibody (magenta) marking the cell junctions and FOP (green) marking the centrioles, scale bar: 10 μm. Bottom right the image segmentation with stem cells colored in gray. **b** Reconstruction of **a** using SET, stem cells in gray. **c** Count of the stem cell to stem cell contacts in the reconstruction by SET panel **b** (dotted green), in the segmentation panel **a** (dotted black), and null distributions of the same feature as generated with three methods including ours: SET where cells are randomly moved 1000 times (in blue), a model where stem cells are shuffled over a hexagonal grid (in red), and a model where stem cells are shuffled over the actual segmentation (in black) ("Methods" and Supplementary Fig. 6). The p value of the reconstruction by SET relative to the null distribution obtained with a 1000 random SET is 0.001. **d**, **f**, **h** Evaluation of the three models against known synthesized patterns all produced from image **a**. **d**, **e** Synthesized clustered stem cells. Our approach by random SET assesses that pattern **d** is clustered as it safely rejects the null hypothesis with a p value ≤ 0.001 (green line outside our null distribution), the two other models consider pattern **d** as random while it is obviously clustered. **f**, **g** Synthesized randomly introduced stem cells. Our approach by random SET cannot reject the null hypothesis (green line inside our null distribution), the two other models consider pattern **f** as unclustered (stem cells would repulse each other). **h**, **i** Synthesized repulsed stem cells (anti-clustering). Unclustered/repulsed stem cells are detected as unclustered by the three models. N = 1 image.

cell types. *Xenopus*' mucociliary epidermis is made of four different cell types at larva stage: multiciliated cell (MCC), goblet cell (GC), small secretory cell (SSC), and ion-secreting cell (ISC)[21–23] (Fig. 4a, c). Differentiation of MCC, SSC, and ISC are driven by the delta-notch pathway under the epithelial layer and are then intercalated in the epithelium from the tissue beneath. These mechanisms end in a spatial MCC organization that was shown to exhibit a stereotypical spacing pattern[24] (Fig. 4a). Spreading between cells of the same type, such that they are rarely in contact with one another, has been visually observed for a long time[25]. The number of contacts with other intercalating cell types was quantitatively measured for ciliated cells and intercalating non-ciliated (INC) cells by Stubbs et al.[26] in 2006 on a thousand cells. Stubbs et al. found that multiciliated cells were generally

not in contact with one another in the apical part of the epidermis at a late embryonic stage. However, other types of INC cells could be in contact both with one another and with ciliated cells.

These spacing rules observed between cells of the same type are not clearly explained. Are they the consequence of an inhibition pathway followed by the intercalation process as suggested by the literature or are they simply the consequence of mechanical effects driven by the shape difference between cell types? In any case, the cell intercalation leads to a difference in shape between intercalating cell types[26]. For instance, from Stubbs et al, the INC seem to be more columnar than elliptical. Altogether, the cell shape variation could explain the limited contacts between small cells.

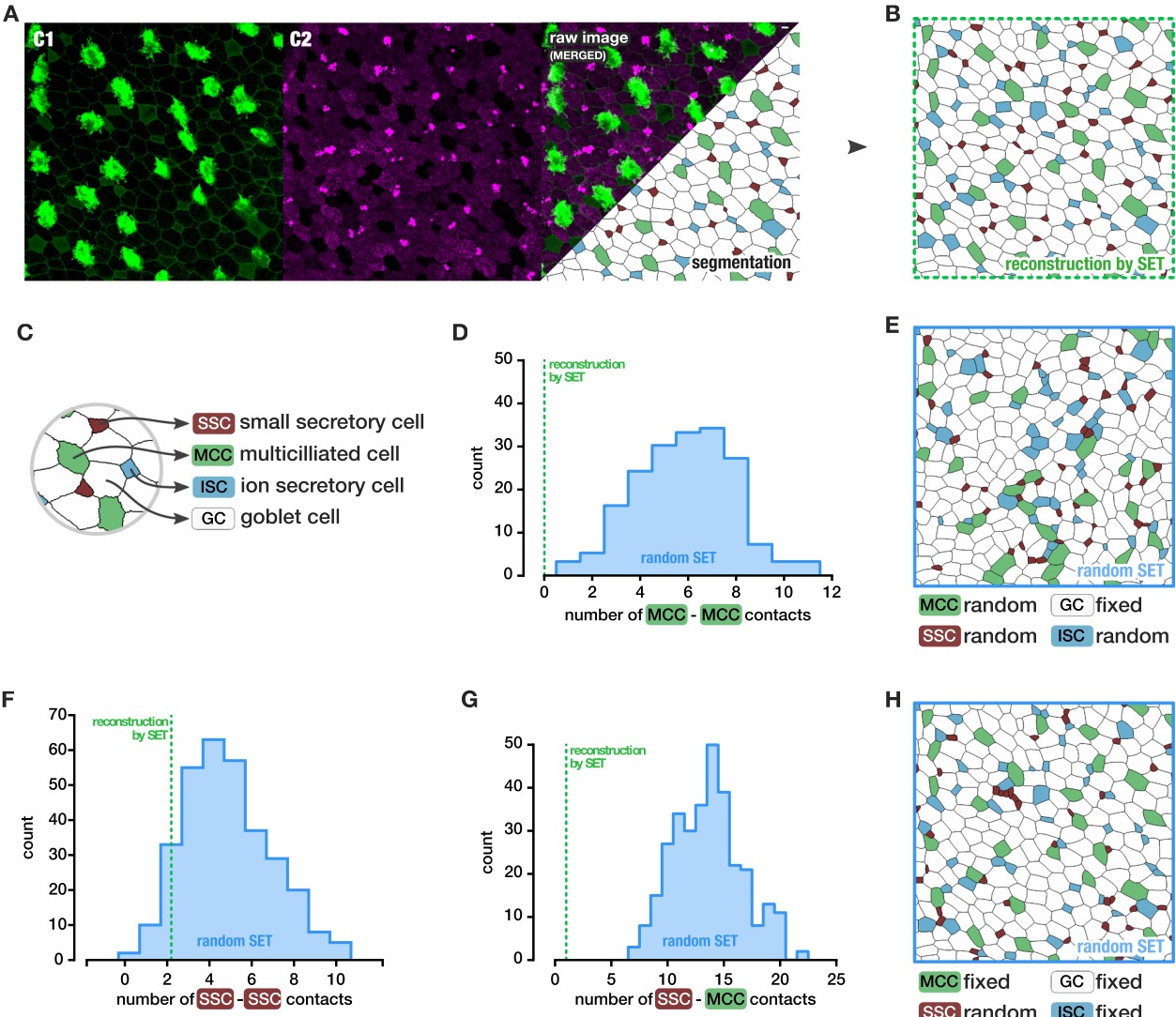

**Fig. 4 Cellular organization of the *Xenopus* tailbud embryo epithelium. a** Image of a st. 33 *Xenopus* epithelium, actin in membrane and cilia labeled with phalloidin-488 and acetylated α-tubulin-488 (C1, green). Lectin PNA 594-stained mucins to define the four cell types[47] (C2, magenta). Scale bar: 10 μm. Segmentation and labeling of four cell types: Multiciliated cells (MCC) are large cells with a high level of actin stain and a lack of lectin PNA. Goblet cells are large cells containing mucins (lectin PNA) and a lack of apical actin. Ionocyte (ISC) are small cells with visible actin and a lack of lectine PNA. Small secretory cells (SSC) are characterized by the absence of actin and dotted lectin PNA highly stained. **b** Reconstruction of image **a** by SET, with cell type preserved. **c** Legend of cell types. **d** MCC–MCC contacts count observed in the reconstruction by SET (green) compared to the counts obtained in a distribution of SET with random position of the intercalating cells only (blue distribution): *p* value < 0.006. **e** An example of SET with random position of the intercalating cells. **f** SSC–SSC contacts count observed in the reconstruction by SET (green) compared to the counts obtained in a distribution of SET with random position of the SSC only (blue distribution): *p* value = 0.141. **g** MCC–SSC contacts count observed in the reconstruction by SET (green) compared to the counts obtained in a distribution of SET with random position of the SSC only (blue distribution): *p* value < 0.003. **h** An example of SET where only the small secretory cells positions are reshuffled. **e** An example of SET with random position of the SSC only. *N* = 1 image.

Using our approach, we generated distributions of alternative SET patterns of intercalations of the *Xenopus* epidermis at stage 33 (Fig. 4a). All intercalating cell types were randomly positioned (all cells except the goblet cells) (Fig. 4e). We could then extract the number of connected MCC from each of those random SET to build the null distribution corresponding to the hypothesis stating that the process of intercalation is random. None of hundreds random SET could produce an image with no MCC connected (Fig. 4d) confirming observation that was primarily reported (Fig. 4b). The probability of observing at least this spacing between MCC by chance was under 0.006. We could then confirm that the MCC positioning process is not random, suggesting that a process promotes MCC spacing.

SSC were characterized more recently and their spatial organization has not been described yet. In (Fig. 4b) only two SSC–SSC contacts can be observed. As the positioning by intercalation of SSC arises later than the MCC intercalation[21,22], we also wondered if the actual sequence of intercalation could explain the small amount of contacts between SSC by randomly shuffling only the SSC. To this aim, we generated random SET such as to fix the positions of all cells except the SSC prior random intercalation of the SSC in a tissue layer made of goblet cells, MCC and ISC (Fig. 4h). In this case, in opposition to MCC, a random spatial organization could not be excluded with a *p* value of 0.141 (Fig. 4f and Supplementary Fig. 9a)

Furthermore, SSC was described as influencing MCC beating with serotonin secretion[21]. Very few contacts could be found

between SSC and MCC in Fig. 4b. With the same random intercalation of SSC, a distribution of random SET could not reproduce the small amount of contact between MCC and SSC with a p value of 0.001 (Fig. 4g and Supplementary Fig. 9B). We then suggest that the spreading process between MCC and SSC is also guided by a deterministic process.

**SET reveals cell neighborhood subcellular organization**. We wanted to test if the location of an organelle inside a cell could be directly or indirectly related to the location of the same organelle in neighboring cells and beyond. At embryonic stages, the ventricular zone of the brain is composed of radial glial cells extending a primary cilium from the apical contact of the cell toward the brain ventricle (Fig. 5a). It was shown that these cells have a translational polarity[27,28]. Indeed, measures of local spatial polarity could be processed but spatial polarity as such is complex to prove statistically because of the heterogeneity across the tissue and the biological variability between samples. Concretely, planar polarity is not homogeneous at the global scale and must be investigated with regard to the local neighborhood (Supplementary Fig. 10A). Importantly, in the absence of a relevant mutant or control, whatever measure is finally considered, it should be related again to a null distribution describing the extent of what can be expected in case there is no local spatial polarity. As mentioned earlier, we included an option that enables to transport the content of each cell from the original image to the reconstruction by SET or to any random SET (Fig. 5b). It allows to preserve the centriole positions location in each cell, as its unknown distribution cannot be considered uniform (Supplementary Fig. 10c, d). We could further show that distribution of intracellular features such as the distance between the center of the cell and the centriole inside each cell was preserved by this process in both the reconstruction and the random SET (Fig. 5c, top). Furthermore, the vector that goes from the centroid of the cell to the centriole was retrieved from each cell. The mean absolute angle formed by this vector on each cell and the same vector in all neighboring cells was then computed (Fig. 5d). We could show that this distribution was preserved for the reconstructed SET while it was expectedly broken in the random SET (Fig. 5c bottom). The same angle could be computed in growing perimeters of neighborhood sorted by rank (distance in cells to the considered cell) (Fig. 5d). This value was then averaged per rank on the whole image to provide one mean absolute angle value per synthetic image. The null distributions we obtained from random SET for the first three ranks were centered on a 90° angle (exactly 89.96, 89.94, and 89.93) with a decreasing standard deviation (1.093, 0.755, and 0.639) (Fig. 5e). Furthermore, as the value extracted per image was the mean of roughly independent angles (most couple of cells are far apart from one another), the Central Limit Theorem (CLT) applied and, accordingly, the null distribution obtained with 1000 random SET could be precisely fit by a Gaussian. The parameters of this Gaussian could be estimated and a p value indicating the probability of observing a polarity measured as at least equal to the one computed on the reconstruction by SET could then be precisely processed and was $1.94 \times 10^{-16}$ for the first rank. Note that as the CLT applies, an estimate of the Gaussian parameters could possibly be obtained with only one random SET. Supplementary Fig. 10E shows that the conclusion drawn from a single random SET this way would be similar and represent a significant gain in computation time. With only one image and without the need of a mutant, this result validates that a mechanism tends to orient the centrioles of close cells in the same direction which is barely discernible on the original image visually as well as quantitatively (Supplementary Fig. 10A, B). We could further show that at larger rank of

neighboring cells, the mean angles of the reconstructed SET keeps on displaying a statistically significant planar polarity with an intensity that decreases with the rank (Fig. 5e). In short, the closer the cells are the more similar their centroid-centriole vector orientation is.

## Discussion

In this work, we introduced SET, a method that reconstructs the cell tessellation of an image as well as random synthetic alternative tessellations made of the exact same cells. We demonstrated that SET reconstructs accurately various tissue types and preserves cell size and shape after shuffling. We then used SET to produce a null distribution to test the significance of an observed pattern and illustrate it on several examples.

The relevance of a statistical test directly relies on the exactness of its null distribution. We showed in the results that the two alternative approaches, resampling over a hexagonal grid or over the actual cell segmentation were biased and thus probably produced inaccurate null distributions. SET simulations illustrate that variation of cell size alone (with constant shape ratio) or shape ratio alone (with almost constant size) lead to variable cell contact distributions, thus showing a relationship (Supplementary Fig. 7). It also shows that the simultaneous variation of both features, or further combined with cell asymmetry, produces skewed and multimodal null distributions of neighbors counts that would be hard to predict given an observed dataset. Indeed, it would require to derive a parametric model of this relationship (if it exists and if it is at all possible) for which parameters could be accurately estimated from one image. On the other hand, by disregarding this existing relationship between cell size and shape, and cell neighbor count, alternative approaches necessarily lead to the construction of an incorrect null distribution in general (Supplementary Fig. 8). These methods would work only in the case where cells are organized as a honeycomb grid, a hypothesis rarely met. SET takes into account cell shape, size, and asymmetry to accurately sample such a null distribution without the need to actually derive a parametric model.

The SET method presents several advantages. We showed that it could be used to assess the existence of a local spatial distribution of cells but also the existence of a relationship in the relative distribution of organelles within neighboring cells, which is particularly subtle and thus hardly perceptible by the human eye. Importantly, only one image is needed to produce a p value, meaning that any location can be investigated independently across a large slide of tissue. Notably, there is no need for parameter tuning as there are no input parameters to the method. The method takes as input an image of segmented cells and all single-cell parameters are automatically estimated when fitting individual distance functions to each cell. It is useful to use a computing cluster to generate several random SET in order to build a null distribution for any given feature. However, this step can be avoided by choosing the feature to be the mean over the image of a local event. In this case the CLT applies and a single random SET is sufficient to compute an estimation of the mean and standard deviation of the sample mean normal distribution parameters and derive a p value as demonstrated in Supplementary Fig. 10E.

A limitation of our method is that it requires the entire image to be occupied by cells. Therefore, it is suited to work on tissue or on cell culture where confluence is 100%. Ultimately, when cells do not occupy all the space, cropped regions can be considered. Note that while the cell relationships in culture can also be interesting to quantify, we anticipate that it is anyway more relevant to tissue samples displaying patterns found in organisms. Another limitation is that while the eight parameters distance

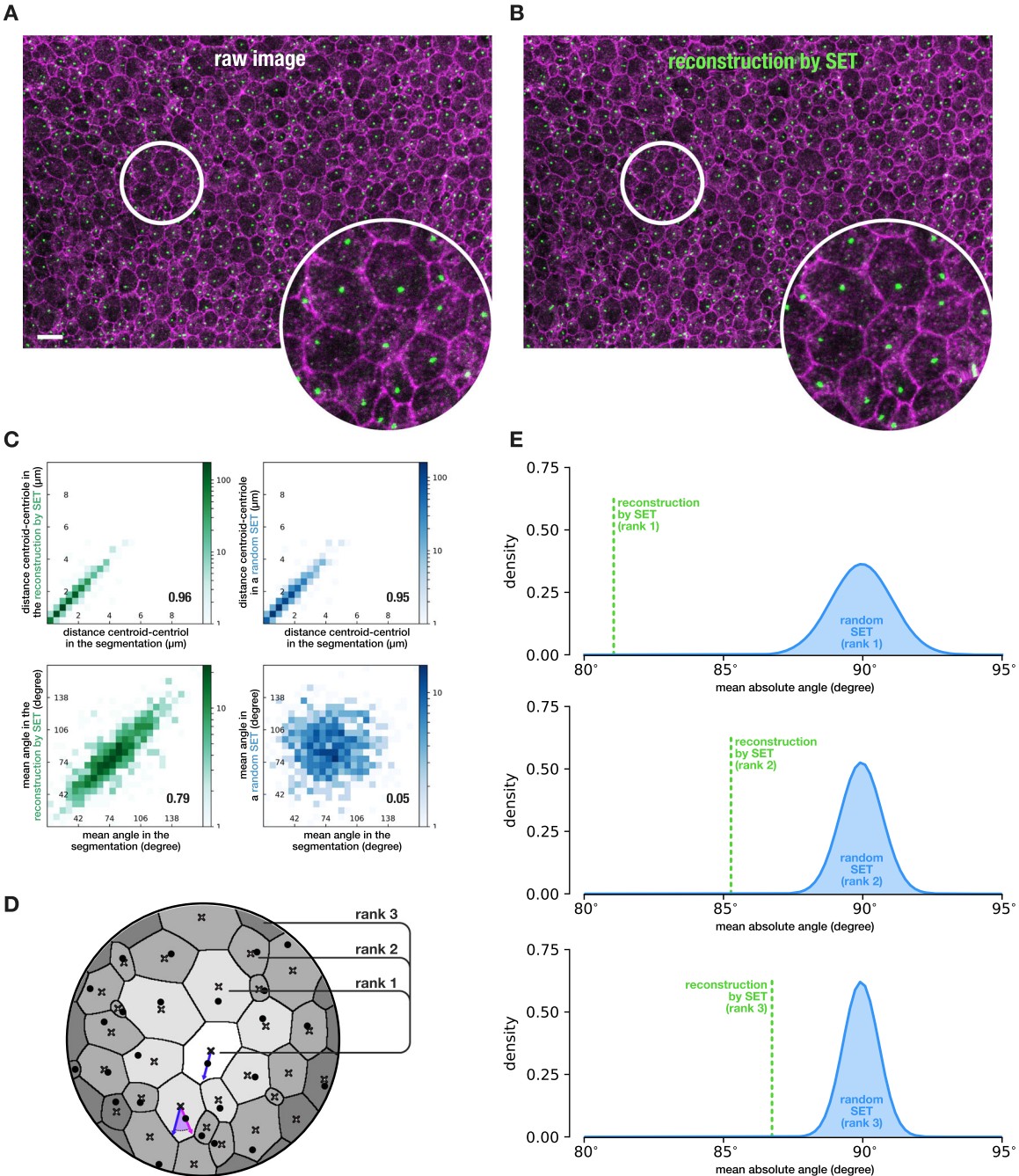

**Fig. 5 Planar polarity organization of the centrioles in E18 mouse ependyma. a** Crop of the maximum intensity projection from a stack of E18 mice ventricular wall with junction labeled with zo1 and centrioles labeled with centrin and ninein, scale bar: 10 μm (full image in Supplementary Fig. 10). **b** Reconstruction of image **a** by SET. Additionally, the content of each cell was morphed from its original location in **a** to the corresponding reconstructed cell in **b** using barycentric coordinates so as to obtain the new location of the centrioles in random SET. **c** Content morphing validation. Quantitative features computed on the content of each cell were preserved after morphing both on the reconstruction by SET and on random SET. Furthermore, quantitative features computed on the content of each cell relative to the neighbor cells were expectedly preserved after morphing on the reconstruction by SET and expectedly lost on random SET. Correlations are Pearson correlation. **d** Description of the quantitative feature considered. Crosses denote cell centroids while dots denote the location of the centriole in each cell (this location can be retrieved in any SET thanks to the morphing step). The vector from the centroid to the centriole is thus defined for all cells. The angle between this vector in the light gray cell (in blue) and the same vector in all the other cells of a given rank (in purple) is computed and averaged by rank to obtain a mean angle value per cell per rank. This value is then averaged per image. **e** Distributions of the value described in **d** for random SET (in blue) and for the reconstruction by SET (in green) both from image **a** for ranks 1 to 3. The distribution of mean absolute angles was approximated by a Gaussian density thanks to the Central Limit Theorem and $p$ values for the reconstructed images by SET were computed: $1.94 \times 10^{-16}$ for rank 1, $3.057 \times 10^{-10}$ for rank 2, and $3.021 \times 10^{-07}$ for rank 3. $N = 1$ image.

function we designed can model a large variation of cell shapes as we demonstrated in the results, some highly non-convex cell shapes will still be hard to fit this way. This is for instance the case for puzzle-shaped cells present in plant leaves[29–31].

The SET model we proposed is used here to shuffle the position of all cells in an image in order to build a statistical test. However, we anticipate that this model could be extended and used in other ways. First, SET is directly applicable to three-dimensional (3D) image stack to analyze in depth tissues[32]. The limitation concerning 3D applications does not come from the proposed methodology but from the fact that the point spread function of fluorescence microscopes are highly anisotrope. Together, with a lower sampling on the axial axis, this often degrades the quality of the top and bottom cell membrane marker such that 3D cell detection, prior to the computation of SET, is in practice rather imprecise. We also envision that the model could be used for biophysical modeling. In this study we modified only the positional parameters in random SET, maintaining all cell shape parameters constant. However, all the parameters for each cell can be modified dynamically. In this way, the SET method could directly be used in an elegant fashion to model cell growth, migration, mitosis, extrusion, and other events to study tissue formation[33,34]. The distance function itself could also be improved to actually model cell pressure on a chosen direction. It would also permit us to model the dynamic of cellulosic walls such as those we can find in some plant cells[35].

Altogether, we introduced the SET model that parametrizes a cell tissue. Furthermore, we illustrate a powerful way to use it to decipher, from a single image and no input parameters, whether any observed quantitative cell relationship feature could be considered as random or not. We anticipate that this approach to modeling tissue has the potential to cover a broad range of applications in computational biology, from biophysical interaction to cell morphodynamic.

## Methods

**Principle of the method**. The core of the SET method relies on approximating the segmentation of each single cell by the level 1 of a flexible parametric distance map. These parametric distance functions, thus independently defined for each single cell, will subsequently be used by a Lloyd-like algorithm to iteratively assign pixels of the plane to the closest cell until convergence. To this aim, we designed a specific and flexible parametric distance function such as to fit a large panel of cell shape. In short, a distance function is designed such that a pixel at distance 1 from the center of the cell should be located on the cell border, a pixel at a smaller distance than 1 should be located inside the cell, and a pixel at a greater distance than 1 should be located outside the cell. The parametric function we designed sums up to eight parameters per cell. These parameters are all estimated once for each cell at the beginning of the process and five of them, describing the shape of the cell, are then fixed to force boundaries of each cell, in order to maintain the same shape in a smooth way. The three remaining parameters describing the location and the orientation of each cell are re-estimated at each iteration of the Lloyd-like algorithm, leaving the cells free to move or rotate until convergence. We show that the overall cell tessellation obtained with various tissues can be accurately reconstructed by SET with only eight parameters per cell using this approach (Fig. 1). We further show that modifying the initial set of location and orientation parameters while still keeping constant the shape parameters enables the synthesis of thousands of alternative tessellations from the same cells (Fig. 2). Further details on the design of the parametric distance, the identification of the parameters by fitting to a cell contour and the reconstruction of the tessellation, are provided in the sections below. We also describe how the content of each cell can be preserved in the synthetic images and how statistical significance of any observed cell pattern can be obtained using SET.

**Design of a flexible parametric distance**. Let $\mathbf{y}$ be a bivariate random vector following a centered standard joint uncorrelated (not necessarily normal) distribution such that $E(\mathbf{y}) = 0$ and $cov(\mathbf{y}) = \mathbf{I}$, $\mathbf{S}$ a diagonal scaling matrix, $\mathbf{R}$ a rotation matrix, and $\boldsymbol{\mu}$ a translating vector. Scaling, rotating, and translating $\mathbf{y}$ yield $\mathbf{x} = \mathbf{RSy} + \boldsymbol{\mu}$. Similarly, $\mathbf{y}$ can be retrieved from $\mathbf{x}$ by inverting the transformation $\mathbf{y} = (\mathbf{RS})^{-1}(\mathbf{x}-\boldsymbol{\mu})$. It is then straightforward to show that $E(\mathbf{x}) = \boldsymbol{\mu}$, $cov(\mathbf{x}) = \mathbf{RSSR'}$ and to retrieve that the Euclidean distance between the origin and $\mathbf{y}$ is the Mahalanobis distance (parameterized by $\boldsymbol{\mu}$, $\mathbf{R}$, and $\mathbf{S}$) between the origin and $\mathbf{x}$

which is the scaled, rotated, and translated vector $\mathbf{y}$:

$$d_{L_2}(0,\mathbf{y}) = \sqrt{\mathbf{y}'\mathbf{y}} = \sqrt{(\mathbf{x}-\boldsymbol{\mu})'(\mathbf{RSSR'})^{-1}(\mathbf{x}-\boldsymbol{\mu})} = d_{M(\boldsymbol{\mu},\mathbf{R},\mathbf{S})}(\mathbf{x}).$$

Independently, the Euclidean distance $d_{L2}$ can be rewritten in the following uncommon way:

$$d_{L_2}(0,\mathbf{y}) = \sqrt{\mathbf{y}_1^2 + \mathbf{y}_2^2} = (1'\mathbf{y}^{\circ 2})^{\frac{1}{2}},$$

where the symbol $\circ$ means that the exponent 2 is applied to the vector $\mathbf{y}$ elementwise. By simply replacing $\mathbf{y}$, the Mahalanobis distance $d_M$ can then also be rewritten this way:

$$d_{M(\boldsymbol{\mu},\mathbf{R},\mathbf{S})}(\mathbf{x}) = d_{L_2}(0,\mathbf{y}) = \left(1'\left|(\mathbf{RS})^{-1}(\mathbf{x}-\boldsymbol{\mu})\right|^{\circ 2}\right)^{\frac{1}{2}}.$$

Unlike the usual quadratic form of the Mahalanobis distance showed earlier, this form presents the compelling advantage of making possible the generalization of Mahalanobis and Minkowski distances (such as Euclidean, Manhattan and Chebychev) under a single parametric function that we name Minkovski Affine Transform (MAT) distance by introducing a parameter $p$ that denotes the Minkovski order:

$$d_{MAT(\boldsymbol{\mu},\mathbf{R},\mathbf{S},p)}(\mathbf{x}) = d_{L_p}(0,\mathbf{y}) = \left(1'\left|(\mathbf{RS})^{-1}(\mathbf{x}-\boldsymbol{\mu})\right|^{\circ p}\right)^{\frac{1}{p}}.$$

Note that if $p \geq 1$, dMAT is a metric, especially if $p = 2$ the $d_{MAT}$ is the Mahalanobis distance and if $p < 1$, triangle inequality is lost and $d_{MAT}$ is a semi-metric. This formulation offers the possibility to design flexible distance functions based on the affine transformation of any Minkovski metric. This relationship between original Minkovski metrics and their transformation to a parameterized MAT distance function are illustrated by Supplementary Fig. 11.

The level sets of MAT are super ellipses that are more flexible than the standard ellipses provided by the Mahalanobis distance. They offer the possibility of modeling roundish rectangular shapes such as some plant cells or diamond like cells. However, they are all symmetric about their center and about the two principal axes. In order to obtain a distance function with level sets possibly matching asymmetric cell shapes, we generalized the MAT distance further by introducing two asymmetric terms $a_1$ and $a_2$. These terms weight how much the value of an axis influences the value of the other axis and reversely, considering the yet unrotated and unscaled vector $\mathbf{y}$. The Asymmetric Minkovski Affine Transform (AMAT) distance $d_{AMAT}$ we propose reads:

$$d_{AMAT(\boldsymbol{\mu},\mathbf{R},\mathbf{S},\mathbf{A},p)}(\mathbf{x}) = \left(1'e^{-\mathbf{AJ}\mathrm{diag}(\mathbf{y})\mathbf{J}}|\mathbf{y}|^{\circ p}\right)^{\frac{1}{p}}$$

with

$$\mathbf{A} = \begin{bmatrix} a_1 & 0 \\ 0 & a_2 \end{bmatrix} \quad \mathbf{J} = \begin{bmatrix} 0 & 1 \\ 1 & 0 \end{bmatrix} \quad \text{and} \quad \mathbf{AJ}\mathrm{diag}(\mathbf{y})\mathbf{J} = \begin{bmatrix} a_1y_2 & 0 \\ 0 & a_2y_1 \end{bmatrix}$$

for the sake of clarity we recall here that

$$\boldsymbol{\mu} = \begin{bmatrix} \mu_1 \\ \mu_2 \end{bmatrix} \mathbf{S} = \begin{bmatrix} s_1 & 0 \\ 0 & s_2 \end{bmatrix} \mathbf{R} = \begin{bmatrix} \cos(\alpha) & \sin(\alpha) \\ -\sin(\alpha) & \cos(\alpha) \end{bmatrix} \mathbf{y} = (\mathbf{RS})^{-1}(\mathbf{x}-\boldsymbol{\mu}).$$

Altogether, the AMAT distance comprises eight parameters: $\mu_1$, $\mu_2$ the coordinates of the cell, $s_1$ the length of the longest axis containing $\boldsymbol{\mu}$, $s_2$ the length of the shortest orthogonal axis containing $\boldsymbol{\mu}$, $\alpha$ the angle of the longest axis containing $\boldsymbol{\mu}$ with the x-axis, $a_1$ the degree of asymmetry about the longest axis, $a_2$ the degree of asymmetry about the shortest orthogonal axis, and $p$ the Minkovski order. This function offers a distance map with short range level sets modeling a large panel of closed shapes such as cells can display (Supplementary Fig. 12). Note that, unlike the MAT distance, some combinations of parameters $a_1$, $a_2$, and $p$ may in theory lead to an AMAT map that can contain critical points at other places than the origin, possibly leading to non-closed or disconnected level sets at long range. In short, AMAT is not guaranteed to be a distance for all combinations of parameters. However, we will see that such a situation can easily be handled, as for our modeling purposes we are only interested in short distances defined locally about the cell membrane, that is about distance 1 from the origin, for which AMAT behaves well as expected.

**Fitting $d_{AMAT} = 1$ to a cell contour**. The segmented contour of each cell is subsampled to an arbitrary resolution of $N$ points $\mathbf{x}_i$ regularly spread (typically $N = 100$). The following sum of squared error is then minimized for each cell:

$$\min_{\boldsymbol{\mu},\mathbf{R},\mathbf{S},\mathbf{A},p} \sum_{i=1}^{N} \left( d_{AMAT(\boldsymbol{\mu},\mathbf{R},\mathbf{S},\mathbf{A},p)}(\mathbf{x}_i) - 1 \right)^2.$$

Note that if $a_1 = 0$, $a_2 = 0$, and $p = 2$ are fixed, then no minimization process is needed, as the AMAT distance is the Mahalanobis distance and the location is the centroid of the cell, and the scale and rotation parameters can be obtained by diagonalization of the covariance matrix of the pixels of the cell. If any of $a_1$, $a_2$, or $p$ are let free to evolve then the minimization process is needed for all parameters and these values are instead used for initialization. The eight parameters of the AMAT distance are then initialized to the centroid of the cell for $\mu_1$ and $\mu_2$, the lengths of the principal axes of the cell for $s_1$ and $s_2$, the angle of the principal axis with the x-axis for $\alpha$, $a_1 = 0$, $a_2 = 0$, and $p = 2$. The parameter $p$ enables modeling

of squarish cells, the parameters $a_1$ and $a_2$ enable triangular modeling or egg like cells, and most importantly, combinations of all of the eight parameters enable a large set of complex cell shapes to be modeled. Whether it is arbitrarily decided to fix some known parameters or not, after this fitting, each cell is represented by a vector of eight parameters that describe a specific parametrization of the AMAT distance function. The level 1 of this two-dimensional (2D) function then matches closely the contour of the cell (Supplementary Movies 1–5). For numerical optimization, the L-BFGS-B algorithm available in scipy.optimize was used, as it enabled us to make the process more robust, by introducing some constraints on the range of values the parameters can take.

**Generation of a tessellation from individual cell metrics**. While, for instance, five parameters enable us to model an elliptical shape and eight parameters enables us to model triangular or rectangular shapes, it does not mean that the cell shape will end up being reconstructed exactly as an ellipse, a triangle, or a rectangle. In fact, the competition for space between cells, each equipped with their own distance, will permit reconstruction of the cell pavement accurately, without any holes. To reconstruct the original image tessellation of $K$ cells, we aim at performing the following minimization:

$$\min_{C_1,\dots,C_K} \sum_{j=1}^{K} \sum_{\mathbf{x}_i \in C_j} d_{\mathrm{AMAT}(\boldsymbol{\mu}_j,\mathbf{R}_j,\mathbf{S}_j,\mathbf{A}_j,p_j)}(\mathbf{x}_i),$$

where $C_j$ denotes the set of pixels $\mathbf{x}_i$ that belong to the cell $j$ with $\boldsymbol{\mu}_j$ and $\mathbf{R}_j$ (the location and orientation parameters) left free to evolve while $\mathbf{S}_j$, $\mathbf{A}_j$, and $p_j$ (the shape parameters) are fixed. It can be solved using a modified Lloyd algorithm. Lloyd is usually employed to obtain a Voronoi tessellation of a 2D or a 3D space with the standard Euclidean distance[8,36,37]. In that case all computed distances along the process are similar and do not depend on parameters. Here, we also aim at performing a tessellation but each compartment uses its own parameterized distance function, as described in the previous section, so as to impose the shape of an actual cell. To our knowledge, Lloyd, with a different metric per cell, was not used for modeling cells. Furthermore, the idea of having each of these metrics matching the properties of a real cell is to our knowledge novel. To reconstruct the original image, the first step is similar to Lloyd and consists of computing the distance of all pixels to all cells (using dedicated AMAT distances) and labeling each of those pixels with the label of its closest cell. In the second step, Lloyd was modified such that the location parameters $\mu_1$, $\mu_2$, and $\alpha$ of each cell are updated by minimizing the sum of square error previously described. Those three parameters only are left free to evolve (except for the incomplete cells at the border of the image for which all parameters are fixed). The five other parameters $s_1$, $s_2$, $a_1$, $a_2$, and $p$, describing the cell shape, are estimated once from the original cell segmentations and remain constant along the rest of the iteration process for all cells to maintain their shape. These two steps are repeated until no more pixels change label. At the end of this process, a tessellation that is an accurate approximation of the original cell tessellation is obtained with only eight parameters per cell provided at initialization by fitting (Fig. 1b–d). To synthesize a random tessellation based on all the cells of a given image, the exact same process is used over the same image dimension but the location and orientation parameters are initialized randomly, still keeping the shape parameters for all cells constant. This process applied on the image Supplementary Fig. 1 to produce its reconstruction by SET and three random SET can be visualized in Supplementary Movie 6. Figure 2 shows that random SET preserves single-cell properties of various tissues while expectedly breaking cell relationships.

**Null distribution and associated $p$ value**. It is important to notice that the reconstruction by SET of the original image could possibly be one sample of the random SET, as the construction process is exactly the same. Only the initialization of the positional parameters (location and orientation) willingly differ: for the reconstruction, these parameters are the original one while in synthetic images they are randomly sampled. This is the foundation of the statistical approach we present: a thousand pictures representing alternative random tessellations of the real image are generated and compared to a reconstruction of that real image using the same process. The statistical significance of any quantitative feature computed from a local group of cells can then be obtained the following way. The considered feature is computed on each random tessellation. Altogether, the sample distribution of these values approximates the null distribution of that feature. Then, the computation of that same feature is also performed on the reconstruction of the original image. If the value computed from the reconstruction falls within the null distribution, then by definition the null hypothesis cannot be rejected. If the value obtained is aside from the null distribution, then a $p$ value can directly be obtained as the ratio of random tessellations that display the same or a more extreme value than value computed from the reconstruction of the original image. Note that if the computed feature is a sum or the mean of independent and identically distributed events over the image, as for Fig. 5e, the null distribution can be approximated by a Gaussian under the CLT. The last combines the advantages of obtaining a more precise $p$ value while necessitating in principle the generation of only one random SET.

**Cell texture mapping**. Independent of the reconstruction of the cell tessellation, we additionally transport the texture content of each cell so as to enable the possible statistical analysis of organelle positioning within the context of its cell neighborhood. To this aim we used a particular weighting of barycentric coordinates called the mean value coordinates, developed by Michael S. Floater[38]. The mean value coordinate method offers a way to smoothly morph the content of an arbitrary polygon with the content of another arbitrary polygon with the same number of vertices. As the synthetic cell contour is about the same shape and size as the original cell contour, we do not expect significant distortion of the content if the orientation of the reference coordinates is similar. Therefore, the segmented contour of each cell and its synthetic counterpart were respectively subsampled to an arbitrary resolution of $n$ ordered points $p_i$ and $p_i'$ (typically 100). The first points $p_0$ and $p_0'$ of both contours correspond respectively to the orientation of their major axis so as to align the two shapes. For each pixel of the synthetic shape we then computed the $n$ mean value coordinates relative to the $n$ points of the contour $p_i'$ and applied the same $n$ weights to the $n$ points of the contour $p_i$ to compute a floating point location in the original cell image. A bilinear interpolation of the four closest pixels from that location enabled recovery of a color value that was then used in the synthetic cell (Supplementary Fig. 2). Using this approach, all pixel values of all synthetic cells could be recovered (Figs. 2a, 5b and Supplementary Movie 6).

**Alternative approaches**. Cell compartments constraint the cell centers to be spread from one another, such that they do not behave as freely as points process approach could essentially model. Therefore, regular point process statistics would hardly be relevant for this type of spatial analyses. We then chose to compare our method to two other approaches that could be considered for such analysis (Fig. 3). These two other approaches, like ours, seek to compare the image observation to a null distribution that should capture the variation about the null hypothesis stating that cells are organized randomly. The difference between the three methods essentially lies in how that null distribution is built by computational means and how relevant it is.

**Alternative approach—Shuffle on a hexagonal grid**. The first approach (red distribution Fig. 3) uses a honeycomb grid containing as many hexagonal cells as in the original image[39,40]. For each run, cell identities were assigned randomly with respect to the observed cell type ratio (83 stem cells from a total cell count of 190 cells for Fig. 3) and the number of contacts between two stem cells was retrieved (Supplementary Fig. 6A).

**Alternative approach—Shuffle on the segmentation**. The second approach (gray distribution Fig. 3) uses the segmentation of the original cell pattern and cell identities are shuffled in order to preserve the distribution of the cell shapes while producing a realistic graph of cell adjacency (Supplementary Fig. 6B). In practice, this model produces a null distribution that is close to the one obtained with the honeycomb method (Fig. 3c) and led to close conclusions.

**Raw image information—Mice**. Ependymal images were acquired from E18, P1, and P30 mice. The experiments were performed in conformity with French and European Union regulations and the recommendations of the local ethics committee (Comité d'éthique in experimentation animale no. 005). The date of the vaginal plug was recorded as embryonic day (E) 0.5 and the date of birth as postnatal day (P) 0. Healthy, immunocompetent animals were kept in a 12 h light/ 12 h dark cycle at 22 °C and fed ad libitum. The mice used in this study include OF1 (Charles River Laboratories) and Centrin2-GFP (CB6-Tg(CAG-EGFP/ CETN2)3-4Jgg/J; The Jackson Laboratory).

**Raw image information—Immunostainings**. Wholemounts of the lateral walls of the lateral LV were dissected[27] from animals sacrificed by cervical dislocation and fixed for 15 min in pure methanol at −20 °C. The samples were incubated for 1 h in blocking solution (1× PBS with 0.1% Triton X-100 and 10% fetal bovine serum) at room temperature followed by overnight incubation at 4 °C in the primary antibodies diluted in blocking solution. The primary antibodies used targeted ZO1 (1:100, cell junction marker; Thermo Fischer Scientific), FOP (1:600, centriole marker, Abnova Corporation), Sas6 (1:500, pro-centriole marker, Santa Cruz), -Catenin (1:500, cell junction marker, Millipore). The following day, the samples were stained with species-specific AlexaFluor fluorophore-conjugated secondary antibodies (1:400, Thermo Fischer Scientific or Jackson ImmunoResearch Labs). Nuclei were counterstained with a 1:1500 Hoechst solution (from a 20 mg/ml stock, Sigma-Aldrich), containing the secondary antibodies for 2 h at room temperature.

Finally, the wholemounts were redissected to keep only the thin lateral walls of the LV[20] which were mounted with Fluoromount-G mounting medium (Southern Biotech, 0100-01).

**Raw image information—Others**. The root image is a Col-0 *Arabidopsis thaliana* sample and has been treated by propidium iodide to label cell walls[41] and imaged

with a Zeiss 710 confocal. The root image is a slice from a 3D stack. The shoot apical meristem image is a FM4-64 staining of a Col-0 *Arabidopsis thaliana* and was acquired with a Leica SP2 confocal as described in ref. [42]. The 3D stack was flattened with merryproj[43]. The *Drosophila* image is originally from ref. [44]. The membranes are visualized with antibodies against E-Cadherin. Image of chick basilar papilla is originally from ref. [45] and was recently used in ref. [46]. The samples were treated with anti-cingulin and anti-hair cell antigen to visualize membrane junction and cell identity. The *Xenopus* epidermis image was acquired from a stage 33 larva. The visualization of membranes and cell identities was made possible by phalloidin labeling of the actin, acetylated alpha tubuli-488 for cillia, andlectin-pna-594 to label mucins in goblet cells and SSCs[47]. The image was extracted from a 3D stack using the SME algorithm[48].

**Cell segmentation**. The minimal input to the SET model is an image of segmented cells where each pixel takes as value the integer label that represents all the pixels of the same cell. All 2D cell segmentations presented in this manuscript were performed using a modified version of the "Morphological Segmentation" plugin of the MorphoLibJ package of ImageJ/Fiji[49]. However, this preprocessing step can be performed by numerous other software packages that do exist to segment images of cells. In practice, as only one image is needed, the full automation of the detection process is not required and segmentation can possibly be manually corrected. Prior segmentation, 2D images of *Xenopus* larva epidermis and mice ependyma were extracted from 3D stack using the SME algorithm[48].

**Computational resources**. Lloyld relaxation with hundreds to thousands of cells each equipped with their own metric to iteratively redistribute labels over millions of pixels can be a demanding process. Our approach is faster when using only Five parameters per cell ($xy$ position, rotation angle, and main axes length) as only the covariance matrix of the pixels of each cell need to be computed and the Mahalanobis distance to each cell equipped with its own matrix can be used. The computation of the last is made very efficient by the cdist function from the scipy Python package (see the code for implementation details). Therefore, when the cells could reproduce correctly the observed image using five parameters (e.g. E18 and adult ependymal cells) we choose this option. In this case the approximate computation time was between 1 h 30 min and 10 h for 1000 SET simulations on 200 cpus Intel Xeon Processor 2400 MHz, depending on the image size. When cell were highly asymetrics such that five parameters did not reproduce properly the observation (e.g. *Xenopus*), then we used eight parameters and it took between 9 and 16 days to compute about 300 simulations on the same computer configuration. All calculations were submitted in parallel thanks to the IBENS computing cluster. Note that the code made available offers the possibility to parallelize computation on all CPUs of a single computer. Furthermore, we anticipate that this type of computation would significantly gain to be ported to GPU computing as it can be highly parallelized per cell; however, we have not investigated this possibility.

**Ethical aspect**. The experiments using mice were performed in conformity with French and European Union regulations and the recommendations of the local ethics committee (Comité d'éthique en experimentation animale no. 005). Mice were bred and maintained in the animal facility of IBENS (Agreement 5502 from the French Ministry of Research and Agreement OGM2014 from the Préfecture de Paris-French ministry of interior). The minimal number of animals was used for the project and the procedures implemented ensured their welfare during their lives.

**Reporting summary**. Further information on research design is available in the Nature Research Reporting Summary linked to this article.

## Data availability
The chick image from ref. [46] was provided by David Sprinzak, Guy Richardson, and Richard Goodyear, and was initially from ref. [45], Copyright 1997 Society for Neuroscience. The *Drosophila* image from ref. [44] was provided by Yohanns Bellaiche with the permissions of AAAS. The *Arabidopsis thaliana* root image was provided by Jean-Christophe Palauqui. The *Arabidopsis thaliana* shoot apical meristem image was provided by Katia Belcram. The *Xenopus* image was provided by Peter Walentek. Mice ependyma images were produced by Nathalie Spassky. A copy of these image data is made available on the Github page along with the code to run the method https://github.com/biocompibens/cellmodelling.

## Code availability
The SET method and all necessary images and code to reproduce the results are available as Python scripts from github (https://github.com/biocompibens/cellmodelling). Version v1.0.0 can be found here[50].

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

## Acknowledgements
We thank our colleagues who provided image data: David Sprinzak, Guy Richardson, and Richard Goodyear for the chick basilar papilla; Bellaiche Yohanns for the *Drosophila* dorso thorax; Jean-Christophe Palauqui for the *Arabidopsis thaliana* root; Katia Belcram for the *Arabidopsis thaliana* shoot apical meristem; and Peter Walentek for the *Xenopus* epidermis. We thank Xavier Morin and Mary Ann Letellier for valuable comments on the manuscript and Felipe Delestro from the Bioinformatics platform of IBENS for the design of the figures. This work has received support under the program "Investissements d'Avenir" launched by the French Government and implemented by the ANR, with the references: ANR-10-LABX-54 MEMO LIFE ANR-11-IDEX-0001-02 PSL* Research University.

## Author contributions
A.G. designed and implemented the SET model, E.L. implemented and performed image analysis and all numerical experiments, N.S. performed all lab experiments and microscopy acquisition of mice ependymal images. A.G. and E.L. wrote the manuscript. All authors edited the manuscript.

## Competing interests
The authors declare no competing interests.
