## [Peer Review File · Communications Biology]

Reviewers' comments:

Reviewer #1 (Remarks to the Author):

Review report of the manuscript "Unraveling spatial cellular pattern by computational tissue shuffling"

The problem of identifying spatial intercellular organization patterns in tissues is an important, understudied and one lacking well-defined quantitative measures. In this manuscript, the authors developed a method to assess spatial inter-cellular organization patterns within heterogeneous tissues. The authors presented SET, a method to synthesize cell tessellations from 3 parameters of cell position and 5 parameters of cell shape, and was demonstrate to successfully reconstruct experimental observed tessellations of different tissue types. The authors propose a statistical test to reject the null hypothesis of a random pairwise local cell-cell spatial pattern for pairs of cells from different types within the same tissue. The statistical test is based on random shuffling the cells positions followed by generating a SET tessellation and measuring the simulated versus observed SET-derived number/frequency of adjacent pairs of cells from the type of interest. This method provide statistical significance to reject the null hypothesis of a random spatial pattern from a single input image. The method require no parameter tuning, was demonstrated on two distinct datasets, and in comparison to two alternative approaches. The advantage of the proposed approach stems from constraining the possible randomly-generated spatial organization of cells according to their size and shape to generate a tighter, spatially-constraint, null hypotheses. The authors also demonstrate that intracellular content of cells can be transported to a synthesized cell location and thus used to reveal relations between intracellular organelle organizations in neighboring cells. Last, the authors propose that dynamic modification of cell shape parameters during the SET evolution could be used to study intercellular relations in cellular processes. The manuscript is well written and mostly easy to follow.

The novelty and potential impact of this approach is due to the fact that alternative methods lack the spatial constraints induced by cell shape and size. The authors make this point in the discussion "The main reason is that the size and shape of the cells are not properly preserved in the construction of the null distribution. Altogether, a correct null distribution could not up to now be obtained either empirically using this type of approaches nor analytically because the spatial arrangement of various cell size is difficult to model". This key point is not laid out explicitly prior to the Discussion. I strongly believe that this point and its implications must be more clearly and constructively discussed and demonstrated throughout the manuscript.

- The authors should explicitly show and discuss the correlation between the cell shape features and the number of neighbors which is the main driver behind their bootstrapping statistical approach advantage over the alternative approaches.
- In Figure 3, the stem cells are much smaller than the multiciliated cells. In Figure 3C, the alternative approaches overestimate the number of adjacent stem-stem pairs because the random shuffling assign larger cells (with more neighbors) the "stem" label. This leads to increase in the number of stem-stem cell pairs. According to the same logic, I would expect that analysis of multiciliated-multiciliated cell pairs will lead to reduced number of pairs by the alternative approaches in relation to the random SET results.

- The same ideas can be also demonstrated in the simulated data (Fig. 3D-H) and in the different potential combination of same/different cell types in Figure 4.

- I think that demonstrating these results and explaining them explicitly in the main text is important to help the readers to better follow and appreciate the strengths and limitations of the proposed approach.
- Organelle relative orientation (Figure 5) – is there an advantage over the alternative approaches (or just a simple random shuffling of the cells, without reconstruction, decoupling their spatial location from the analysis)? This would be another important analysis in the same context.

- As a limitation, the authors should explicitly state that their approach is superior to the alternative approaches only in cases of different shape & size distributions between the different cell types. I suggest that beyond explicitly discussing this point in the Discussion section, the authors can also demonstrate this with a simulation.

Another concern is regarding results reproducibility between different replicates. The authors demonstrate their approach with N = 1 image per tissue type. I think it is important to verify the consistency between outcomes across replicates, at least in one of the test cases reported.

Additional comments and suggestions:

- Data availability and reporting N. Please report in the manuscript the N – number of images per tissue type used in this manuscript and make these publically available (I see N = 1 for most test cases, this is not reported in the main text / figure legends / methods).

- The authors mention in the main text alternative approaches that they compare to SET. The results of the application of these approaches are presented in main-text figures but the approaches are not described, even not briefly, in the main text (pointing to online methods). As a reader of the manuscript I would like to understand, at least the intuition behind, the alternative approaches. It would improve readability to include a brief description when these approaches are mentioned in the text.

- Use of technical terms. The authors can improve the readability of their manuscript by providing some intuition of the algorithms that they mention in the main text (in addition to the more detailed description in the Methods). For example “Lloyd algorithm” and “barycentric coordinates”.

- Minor issues:
 - o “let free to evolve and all the pixels in the plan..” ◇ plane
 - o It is very hard to read the blue text (“random SET”) in Fig. 3C.
 - o In figure 3C, E, G, I – frequency should be normalized (so the integral is 1). Otherwise, it should be labeled “count”.

Assaf Zaritsky, Ben-Gurion University of the Negev, Israel

Reviewer #2 (Remarks to the Author):

In this study, Laruelle et al introduce a new method to analyse imaging data from epithelial tissues in a statistically rigorous way. Epithelia are widely studied in the context of physiology, cancer, and morphogenesis. Often, we are interested in understanding spatial heterogeneity of these tissues.

However, if a spatial phenomenon catches our eye, it is impossible to decide whether this phenomenon is a true effect or just a random occurrence that happens to ‘look that way’, and this limitation can truly hamper research progress in many research fields that analyse epithelial patterning. This new method overcomes this challenge by introducing a way to test local patterns for statistical significance. To decide if a certain effect is significant, all cell positions in an image are randomised in a large collection of new mock images. If the effect of interest recurs in these randomised images, then we know it can happen at random. If not, then some mechanism may be at play to achieve this effect. The key achievement of this new method is to perform the randomisation in a way that controls for cell geometry locally and across the tissue. In the new mock images, the cells have similar cell shape and size as before, and tissue properties of the cell network, such as the average number of cell neighbours, are preserved. Even cellular content can be preserved. This method is very original, and I think the way how it is designed is brilliant. One caveat of the method is that cell shapes in the new mock images are not exactly identical to the original cell shapes. The authors are explicit about this, and perform multiple tests to carefully quantify this difference, and show convincingly that these differences are sufficiently small to make the method highly reliable overall. They show the usefulness of their method on multiple examples in often-studied tissue types. The study is well written and easy to follow, despite the necessary technical complexity of the method. I am impressed by this manuscript, I think it will be highly useful to multiple research fields working with epithelial imaging data, and full-heartedly recommend it for publication.

Before publication, I suggest that the authors address the following minor comments:

- 1) It is nice that the method works on single images and I agree that this is what is needed to identify the strength of visible geometrical effects. I appreciate that imaging data is incredibly rich, and that the presence of many cells in single images allows one to perform statistics on them. However, in practice biological replicates are still important. If a result on cell geometry or tissue patterning is observed, one should not draw this conclusion from a single data point, or a single piece of tissue. Can you comment on, or perhaps give guidelines about, how users of this method can integrate this analysis over multiple replicates? Is there anything one can say about how many samples would one need to collect to have confidence in any observed effects?
- 2) In the section where you mention software packages that enable epithelial segmentation, it maybe worthwhile to acknowledge the efforts of many authors who have written new software to achieve this. Examples that come to my mind would be Epitools by Yanlan Mao’s group, Packaging Analyzer by the Juelicher/Eaton labs, Seedwater Segmenter by the Hutson lab, as well as ilastik and CellProfiler. If anything, citing these softwares will give pointers to readers who seek to find the right segmentation software before being able to use your method.
- 3) You are using a version of Lloyd’s relaxation to generate accurate SET reconstructions. The utility of Lloyd’s relaxations in reproducing epithelial cell geometries has previously been identified by some other groups, specifically this one:
<https://www.embopress.org/doi/full/10.15252/emboj.201592374>
They build on previous work describing Epithelia using Voronoi tessellations, for example this one:
<https://www.sciencedirect.com/science/article/abs/pii/S0022519378903156>
It may be worthwhile acknowledging some of that previous work by citing them.
- 4) Regarding calculation times, how long does it take to run this algorithm? You mention that it may be useful to run the code on a research cluster, which suggests that it is computationally expensive. Could you perhaps, for each of the examples you report, include what resources you used to run this, and how long it took, or report typical calculation times in some other way?
- 5) In the polarity problem on figure 5, you run only one SET randomisation, arguing that it is

sufficient to do so as the Central Limit Theorem applies. Have you double checked that this is true by repeating the analysis on a second SET randomisation? Does this give similar p-values?

6) Potential typo: at the beginning of the results section: 'After fitting, this parametric distance function combines height parameters per cell'. Should it be 'eight' parameters instead?

Point by point answers to the reviewers comments

We thanks reviewers for their very constructive comments. We basically agreed with most of them, modified the manuscript accordingly and provided additional numerical experiments as supplementary figures. We answer in details in blue below and modified sections of the manuscript in red accordingly.

Reviewer #1 (Remarks to the Author):

Review report of the manuscript “Unraveling spatial cellular pattern by computational tissue shuffling”

The problem of identifying spatial intercellular organization patterns in tissues is an important, understudied and one lacking well-defined quantitative measures. In this manuscript, the authors developed a method to assess spatial inter-cellular organization patterns within heterogeneous tissues.

The authors presented SET, a method to synthesize cell tessellations from 3 parameters of cell position and 5 parameters of cell shape, and was demonstrate to successfully reconstruct experimental observed tessellations of different tissue types.

The authors propose a statistical test to reject the null hypothesis of a random pairwise local cell-cell spatial pattern for pairs of cells from different types within the same tissue. The statistical test is based on random shuffling the cells positions followed by generating a SET tessellation and measuring the simulated versus observed SET-derived number/frequency of adjacent pairs of cells from the type of interest. This method provide statistical significance to reject the null hypothesis of a random spatial pattern from a single input image. The method require no parameter tuning, was demonstrated on two distinct datasets, and in comparison to two alternative approaches. The advantage of the proposed approach stems from constraining the possible randomly-generated spatial organization of cells according to their size and shape to generate a tighter, spatially-constraint, null hypotheses. The authors also demonstrate that intracellular content of cells can be transported to a synthesized cell location and thus used to reveal relations between intracellular organelle organizations in neighboring cells. Last, the authors propose that dynamic modification of cell shape parameters during the SET evolution could be used to study intercellular relations in cellular processes. The manuscript is well written and mostly easy to follow.

Thanks for appreciating our work.

The novelty and potential impact of this approach is due to the fact that alternative methods lack the spatial constraints induced by cell shape and size. The authors make this point in the

discussion “The main reason is that the size and shape of the cells are not properly preserved in the construction of the null distribution. Altogether, a correct null distribution could not up to now be obtained either empirically using this type of approaches nor analytically because the spatial arrangement of various cell size is difficult to model”. This key point is not laid out explicitly prior to the Discussion. I strongly believe that this point and its implications must be more clearly and constructively discussed and demonstrated throughout the manuscript.

We agree with reviewer #1. It is indeed a key point. Therefore we added a section at the end of the introduction to explicitly mention it at the beginning of the manuscript. We also provide simulations in two additional supplementary figures to answer the specific points mentioned below and modified the manuscript accordingly in results and discussion. Indeed, the aim of this work is to provide SET as a solution to this issue. We hope this is clearer.

- The authors should explicitly show and discuss the correlation between the cell shape features and the number of neighbors which is the main driver behind their bootstrapping statistical approach advantage over the alternative approaches.

We performed additional simulations using SET to construct various types of cell epithelia and explore the variety of derived cell contact distributions. We also display the cell contact distributions of all real image of the manuscript on the same Additional supplementary figure 7. It illustrates that inhomogeneous distribution of cell size alone (with constant shape ratio) or shape ratio alone (with almost constant size) lead to variable cell contact distributions. It also shows that the combination of both, or further combined with cell asymmetry (as we can model with SET), produces skewed and multimodal null distributions of neighbors counts that would be hard to estimate from a given observed dataset. It would require to explore the relationship between the joint distribution of shape features and the distribution of cell neighbors and derive a parametric model (if it is at all possible) for which parameters can be estimated from one image data. This new supplementary figure 7 is first pointed at the end of the results on adult ependyma to give a hint on why result obtained with the two other methods are biased. We also describe it further in the discussion. We cannot present this study earlier in the manuscript as it uses SET that is described in the two first result sections.

- In Figure 3, the stem cells are much smaller than the multiciliated cells. In Figure 3C, the alternative approaches overestimate the number of adjacent stem-stem pairs because the random shuffling assign larger cells (with more neighbors) the “stem” label. This leads to increase in the number of stem-stem cell pairs. According to the same logic, I would expect that analysis of multiciliated-multiciliated cell pairs will lead to reduced number of pairs by the alternative approaches in relation to the random SET results.

We confirm this is the case (additional supplementary figure 4B panel). Furthermore, this is an artefact that is made obvious by some additional simulation we performed displayed in the additional supplementary figure 8. The last show that shuffling labels across two cell populations, ignoring the shape and size specific to each subpopulation, produces a similar

distribution of the number of neighbors for both populations which is incorrect. We point this in the third result section and discuss it in the discussion.

- The same ideas can be also demonstrated in the simulated data (Fig. 3D-H) and in the different potential combination of same/different cell types in Figure 4.

We also have added these additional results in supplementary figure 4 C,D,E

- I think that demonstrating these results and explaining them explicitly in the main text is important to help the readers to better follow and appreciate the strengths and limitations of the proposed approach.

We modified the introduction, the third result section and the discussion and presented additional supplementary figure 4, 7, 8 to this end.

- Organelle relative orientation (Figure 5) – is there an advantage over the alternative approaches (or just a simple random shuffling of the cells, without reconstruction, decoupling their spatial location from the analysis)? This would be another important analysis in the same context.

Good question. It would be relevant to sample for the location of the centriole in each cell independently, if the centriole location distribution was uniform. However, we computed it and it is clearly not (Supplementary figure 10C). Meaning that sampling positions uniformly would produce an unreliable null distribution to compare our observation to and would lead to an irrelevant test. Following this idea, we wondered if it would be then interesting to sample from this actual sample distribution, but it turned out that it would also lead to a wrong null distribution because we also show in the same supplementary figure that this distribution may itself depend on the size and shape parameters. Indeed, we show that producing the same distribution splitting the cells in two by size or by shape ratio, displayed different spatial distributions (Supplementary figure 10D). Shuffling the cells preserving their content, as we propose in the paper, presents the compelling advantage to limit bias in sampling organel position without the need to model its distribution. We added this point in the corresponding result section which indeed justify the use of this approach.

- As a limitation, the authors should explicitly state that their approach is superior to the alternative approaches only in cases of different shape & size distributions between the different cell types. I suggest that beyond explicitly discussing this point in the Discussion section, the authors can also demonstrate this with a simulation.

We agree in principle with reviewer #1 on this point and we even had mentioned it at the end of the third result part in the first version. However, we anticipate that cell size and shape do vary in most real case and the honeycomb grid is a rare exception. We performed additional

simulation showing that disregarding differences between cell types shape and size systematically produces a similar neighbor count distribution for both cell types which is inaccurate and lead to a bias (Additional Sup fig 8). We added some text in the third result part and in the discussion that explicitly discuss these and recall the point you mention.

Another concern is regarding results reproducibility between different replicates. The authors demonstrate their approach with $N = 1$ image per tissue type. I think it is important to verify the consistency between outcomes across replicates, at least in one of the test cases reported.

We agree with this point in principle. It is in general necessary to use replicates to evaluate the extent of a difference. However we see the method as a means to obtain reliable statistics per position without the need of searching for the same location in a replicate individual, because it is often unfeasible. However, we agree that to prove that the method works, it is an interesting point to investigate. We then added results we obtained with a biological replicate, from the experiment displayed on Figure 3, taken in another individual at approximately the same location in the tissue. Results are compiled in an additional supplementary figure and confirm the results obtained in the first instance (Supplementary figure 5).

Additional comments and suggestions:

- Data availability and reporting N . Please report in the manuscript the N – number of images per tissue type used in this manuscript and make these publically available (I see $N = 1$ for most test cases, this is not reported in the main text / figure legends / methods).

We mentioned in the first manuscript that data was made publically available (see the link at the end of the data availability section). All experiments required only one image, we added a “ $N=1$ image” in all figure caption as requested.

- The authors mention in the main text alternative approaches that they compare to SET. The results of the application of these approaches are presented in main-text figures but the approaches are not described, even not briefly, in the main text (pointing to online methods). As a reader of the manuscript I would like to understand, at least the intuition behind, the alternative approaches. It would improve readability to include a brief description when these approaches are mentioned in the text.

We added a short description of the two alternative approaches in the main text to ease reading and still point to online methods to a more precise description.

- Use of technical terms. The authors can improve the readability of their manuscript by providing some intuition of the algorithms that they mention in the main text (in addition to the more detailed description in the Methods). For example “Lloyd algorithm” and “barycentric coordinates”.

We briefly described the purpose of the Lloyd iterations and the barycentric coordinates in the main text where we mentioned them to ease reading.

- Minor issues:

- o “let free to evolve and all the pixels in the plan..” plane

- o It is very hard to read the blue text (“random SET”) in Fig. 3C.

- o In figure 3C, E, G, I – frequency should be normalized (so the integral is 1). Otherwise, it should be labeled “count”.

Thanks, we corrected these

Assaf Zaritsky, Ben-Gurion University of the Negev, Israel

Reviewer #2 (Remarks to the Author):

In this study, Laruelle et al introduce a new method to analyse imaging data from epithelial tissues in a statistically rigorous way. Epithelia are widely studied in the context of physiology, cancer, and morphogenesis. Often, we are interested in understanding spatial heterogeneity of these tissues. However, if a spatial phenomenon catches our eye, it is impossible to decide whether this phenomenon is a true effect or just a random occurrence that happens to ‘look that way’, and this limitation can truly hamper research progress in many research fields that analyse epithelial patterning. This new method overcomes this challenge by introducing a way to test local patterns for statistical significance. To decide if a certain effect is significant, all cell positions in an image are randomised in a large collection new mock images. If the effect of interest recurs in these randomised images, then we know it can happen at random. If not, then some mechanism may be at play to achieve this effect. The key achievement of this new method is to perform the randomisation in a way that controls for cell geometry locally and across the tissue. In the new mock images, the cells have similar cell shape and size as before, and tissue properties of the cell network, such as the average number of cell neighbours, are preserved. Even cellular content can be preserved. This method is very original, and I think the way how it is designed is brilliant. One caveat of the method is that cell shapes in the new mock images are not exactly identical to the original cell shapes. The authors are explicit about this, and perform multiple tests to carefully quantify this difference, and show convincingly that these differences are sufficiently small to make the method highly reliable overall. They show the usefulness of their method on multiple examples in often-studied tissue types. The study is well written and easy to follow, despite the necessary technical complexity of the method. I am impressed by this manuscript, I think it will be highly useful to multiple research fields working with epithelial imaging data, and full-heartedly recommend it for publication.

Thanks for appreciating our work.

Before publication, I suggest that the authors address the following minor comments:

1) It is nice that the method works on single images and I agree that this is what is needed to identify the strength of visible geometrical effects. I appreciate that imaging data is incredibly rich, and that the presence of many cells in single images allows one to perform statistics on them. However, in practice biological replicates are still important. If a result on cell geometry or tissue patterning is observed, one should not draw this conclusion from a single data point, or a single piece of tissue. Can you comment on, or perhaps give guidelines about, how users of this method can integrate this analysis over multiple replicates? Is there anything one can say about how many samples would one need to collect to have confidence in any observed effects?

As mention to a similar question by reviewer #1, we agree with this point in principle: it is in general necessary to use replicates to evaluate the extent of a difference. However we see the method as a means to obtain reliable statistics per position without the need of searching for the same location in a replicate individual, because it is often unfeasible. Another reason is that significance of a pattern can vary across one tissue sample and it is interesting therefore to be able to draw conclusions on a single position from a single image. It then becomes possible to investigate the presence of a pattern along a piece of tissue.

However, we agree that to prove that the method works, it is an interesting point to investigate reproducibility if possible. We then added results we obtained with a biological replicate, from the experiment displayed on Figure 3, taken in another individual, at the same age and at approximately the same location in the tissue. Results presented in the additional Supplementary figure 5 confirm the results obtained in the first instance. We modified the manuscript accordingly and reported this result in the results on adult ependyma.

2) In the section where you mention software packages that enable epithelial segmentation, it maybe worthwhile to acknowledge the efforts of many authors who have written new software to achieve this. Examples that come to my mind would be Epitools by Yanlan Mao's group, Packaging Analyzer by the Juelicher/Eaton labs, Seedwater Segmenter by the Hutson lab, as well as ilastik and CellProfiler. If anything, citing these softwares will give pointers to readers who seek to find the right segmentation software before being able to use your method.

We added these references to the paper in the first results part, describing the SET model.

3) You are using a version of Lloyd's relaxation to generate accurate SET reconstructions. The utility of Lloyd's relaxations in reproducing epithelial cell geometries has previously been identified by some other groups, specifically this one: <https://www.embopress.org/doi/full/10.15252/embj.201592374>
They build on previous work describing Epithelia using Voronoi tessellations, for example this one: <https://www.sciencedirect.com/science/article/abs/pii/S0022519378903156>
It may be worthwhile acknowledging some of that previous work by citing them.

We already cited Sánchez-Gutiérrez et al. in the first manuscript when describing the method. We have added the second reference you suggested.

4) Regarding calculation times, how long does it take to run this algorithm? You mention that it may be useful to run the code on a research cluster, which suggests that it is computationally expensive. Could you perhaps, for each of the examples you report, include what resources you used to run this, and how long it took, or report typical calculation times in some other way?

Good point, thanks. We added a “Computational resources” section in the methods to provide details related to the computation time needed and the resources.

5) In the polarity problem on figure 5, you run only one SET randomisation, arguing that it is sufficient to do so as the Central Limit Theorem applies. Have you double checked that this is true by repeating the analysis on a second SET randomisation? Does this give similar p-values?

Sorry if we were unclear in the description, but we have not run only one SET randomization but a thousands for this figure. The Gaussian displayed on Figure 5 is a Gaussian distribution fitted on the sample mean distribution actually obtained from a thousands random SET. We meant to say that the distribution is approximately Gaussian thanks to the Central Limit Theorem and that therefore the distribution can be fit by a Gaussian function. We modified the text to make this clearer. Furthermore, because your point is relevant and refer to a point we had suggested in the discussion and in the method sections, we also verified that approximating the Gaussian parameters using only one of these random SET would lead to the same conclusion. We added the results obtained by any five (randomly chosen!) random SETs in a table in Supplementary figure 10E. They indeed confirm that for a mean value feature, only one random SET is needed which is a significant saving in computation time.

6) Potential typo: at the beginning of the results section: ‘After fitting, this parametric distance function combines height parameters per cell’. Should it be ‘eight’ parameters instead?

We corrected it

REVIEWERS' COMMENTS:

Reviewer #1 (Remarks to the Author):

The authors have addressed my concerns and made this creative, elegant and well-written manuscript, even stronger. The paper should be accepted for publication.

One suggestion, that the authors can decide to incorporate in their manuscript, or not. I think that explicitly showing scatter plots (or joint distribution) of cell shape feature and the number of neighbors could improve the readers' intuition regarding the main driver behind the presented bootstrapping statistical approach.

Assaf Zaritsky, Ben-Gurion University of the Negev, Israel

Reviewer #2 (Remarks to the Author):

In the revised manuscript and the rebuttal letter the authors have addressed my already minor comments, and further illustrated the strengths and utility of their new method to identify cellular patterns in microscopy images of epithelial tissues. I have no remaining concerns and recommend the paper for publication.

I repeat that this method of identifying tissue properties with statistical significance is truly novel and solves a bottleneck in the quantitative analysis of epithelial tissues. The manuscript is well-written and illustrates the accuracy and utility of the method on multiple examples. This method will be a highly useful tool for the various scientific communities working with epithelial imaging.

Jochen Kursawe (University of St Andrews)